# Enhancing Concept Localization in CLIP-based Concept Bottleneck Models

**Rémi Kazmierczak**  *remi.kazmierczak@ensta-paris.fr*
*Unité d'Informatique et d'Ingénierie des Systèmes*
*ENSTA Paris, Institut Polytechnique de Paris*

**Steve Azzolin**  *steve.azzolin@unitn.it*
*Department of Information Engineering and Computer Science*
*University of Trento*

**Eloïse Berthier**  *eloise.berthier@ensta-paris.fr*
*Unité d'Informatique et d'Ingénierie des Systèmes*
*ENSTA Paris, Institut Polytechnique de Paris*

**Goran Frehse**  *goran.frehse@ensta-paris.fr*
*Unité d'Informatique et d'Ingénierie des Systèmes*
*ENSTA Paris, Institut Polytechnique de Paris*

**Gianni Franchi**  *gianni.franchi@ensta-paris.fr*
*Unité d'Informatique et d'Ingénierie des Systèmes*
*ENSTA Paris, Institut Polytechnique de Paris*

**Reviewed on OpenReview:** *https://openreview.net/forum?id=2xaOl0wluw*

## Abstract

This paper addresses explainable AI (XAI) through the lens of Concept Bottleneck Models (CBMs) that do not require explicit concept annotations, relying instead on concepts extracted using CLIP in a zero-shot manner. We show that CLIP, which is central in these techniques, is prone to concept hallucination—incorrectly predicting the presence or absence of concepts within an image in scenarios used in numerous CBMs, hence undermining the faithfulness of explanations. To mitigate this issue, we introduce Concept Hallucination Inhibition via Localized Interpretability (CHILI), a technique that proposes a disentangling of image embeddings. Furthermore, our approach supports the generation of saliency-based explanations that are more interpretable.

## 1 Introduction

Deep Neural Networks (DNNs) are now used in many areas, including sensitive domains such as medicine and law. In these settings, trust is essential. To build trust, the field of Explainable Artificial Intelligence (XAI) provides tools that help users understand how DNNs make decisions. One important family of methods is *concept-based explanations*. These explanations describe predictions using human-understandable concepts, often expressed as words. For example, a model that classifies an image as a *dog* might rely on concepts such as *fur*, *ears*, *snout*, or *paws*. The ability of a model to represent raw data (e.g., images) as concepts—called *conceptual representation*—is therefore key to creating models that can provide such explanations.

A popular way to use concepts is to embed them directly into the model. This creates an interpretable latent space, where each neuron corresponds to a concept. Models built this way are known as *Concept Bottleneck*

*Models (CBMs)* (Koh et al., 2020; Bennetot et al., 2022). While CBMs improve interpretability by design, they usually require concept annotations during training, which are expensive and difficult to collect.

Recently, contrastive language-image models, such as CLIP (Yan et al., 2023a), have been widely used for tasks like zero-shot classification and open-world recognition. Because CLIP links images and text, researchers have started using it as a free source of concepts for CBMs (Yang et al., 2023; Panousis et al., 2023; Cui et al., 2023). This removes the need for manual annotations, but also introduces a new challenge: the concepts extracted by CLIP may not always reflect what is actually in the image.

A particularly critical challenge for CBMs is the phenomenon of *concept hallucination* (illustrated in Figure 1), wherein concepts are inferred based on contextual cues rather than their actual presence within the image. This issue undermines the foundational hypothesis of CBMs—that the concept bottleneck serves as a faithful conceptual representation of the image content. While prior work has addressed related challenges (Oh & Hwang, 2025; Liu et al., 2024b), our approach distinguishes itself in two key aspects. First, we not only mitigate concept hallucinations but also enhance their localization by explicitly addressing the spatial distribution of activation maps. Second, we extend the applicability of our method to CBMs, thereby offering a tailored solution for improving both the reliability and interpretability of these models.

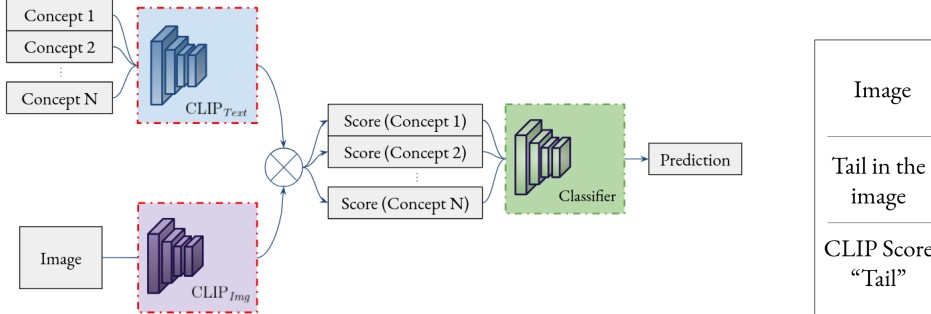 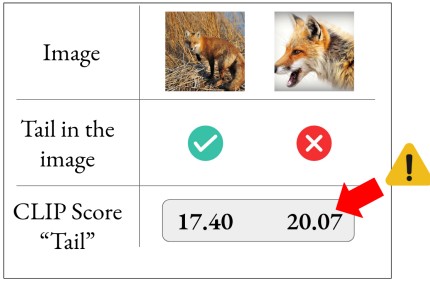

(a) **Principle of a CLIP-based concept bottleneck model.**  (b) **CLIP hallucination issue.**

Figure 1: Using the CLIP-score between embedings of input images and predefined concepts, labeling-free concept extraction can be performed, allowing prediction on an interpretable latent space (left). However, CLIP tends to hallucinate the presence of concepts, troubling the localisation of CLIP-based CBMs (right).

Our contributions are as follows:

- We conduct an extensive statistical analysis to investigate the relationship between the CLIP-score and the localization of concepts. Notably, our findings demonstrate that CLIP-scores fail to accurately represent the actual location of concepts within images.

- Based on this observation, we propose CHILI (Concept Hallucination Inhibition via Localized Interpretability), a method that aims to disentangle the activations of CLIP, and consequently the CLIP-score, distinguishing between object representation, which is related to the physical location of the concept, and contextual representation, which pertains to activations associated with features that do not directly represent the concept but suggest its presence.

- To demonstrate the efficacy of our disentangling method, we employ it as a means to perform image segmentation and binary classification of concepts in spurious situations. We showcase that our method achieves slightly superior results compared to concurrent methods in both tasks.

- We apply CHILI to real-world use cases to construct new, more interpretable CBMs. Our results demonstrate that such an intervention is feasible with only a limited accuracy cost.

## 2 Related Work

**Concept Bottleneck models (CBMs)**  CBMs constitute a class of models that exploit a conceptual representation of input data, termed the "concept bottleneck," to facilitate inference, thereby enhancing interpretability. While certain studies employ custom datasets featuring concept annotations to construct CBMs (Koh et al., 2020; Díaz-Rodríguez et al., 2022), the emergence of text-image contrastive foundation models has significantly propelled research in this area, enabling the development of CBMs without explicit concept annotations (Yan et al., 2023a; Kazmierczak et al., 2024). Notably, CLIP (Yan et al., 2023a) has become the predominant choice for crafting these CBMs (Kazmierczak et al., 2025).

**Neuron interpretation**  To interpret the behavior of a model post-training, a commonly employed approach involves identifying the role of specific neurons in the process by detecting patterns that induce their activation. Some methods directly display neuron activations in response to designed inputs (Gandelsman et al., 2023). More sophisticated techniques use statistical analysis on a probing dataset to achieve this goal (Shaham et al., 2024; Kalibhat et al., 2023). Alternatively, optimization techniques can be employed to determine the input that maximizes the activation of a given neuron (Olah et al., 2017).

**Saliency based explanations**  To explain image-based decisions, saliency-based explanations—which aim to highlight the most influential regions according to the model—are widely adopted. Among these, model-agnostic approaches such as SHAP (Lundberg & Lee, 2017), LIME (Ribeiro et al., 2016), and RISE (Petsiuk et al., 2018) are particularly popular due to their versatility. These methods analyze model behavior in response to perturbed versions of the input image.

Additionally, the formal structure of deep learning models has enabled alternative approaches for generating saliency maps by directly examining activation patterns (Zhou et al., 2016; Gandelsman et al., 2023). Another distinct class of deep neural network (DNN)-based saliency methods leverages gradients to weight activations, as seen in Grad-CAM (Selvaraju et al., 2017), FullGrad (Srinivas & Fleuret, 2019), and HiResCAM (Draelos & Carin, 2020). However, such gradient-based techniques require differentiable computations, a constraint that does not apply to conceptual representations.

## 3 Evaluating the concept localization abilities of CLIP

### 3.1 Preliminaries

First, let us define some notions that we consider essential to describe the experiments we will perform.

**Related studies**  The widespread success of CLIP has spurred significant research effort around its interpretability. Existing studies primarily focus on two issues: bias and spurious feature reliance, and concept hallucination.

The most extensively studied aspect is bias, often examined through image classification tasks. Works such as those by Moayeri et al. (2023b); Zhang et al. (2024) demonstrate accuracy drops on biased datasets, revealing CLIP's reliance on spurious features. Furthermore, Birhane et al. (2021); Hall et al. (2023) show that these biases extend to societal concerns, including gender and racial discrimination. Mitigation approaches include fine-tuning (Alabdulmohsin et al., 2024; Gerych et al., 2024) and activation decomposition (Yeo et al., 2025). Another key challenge is CLIP's tendency to hallucinate text or objects during text-image similarity computations (Oh & Hwang, 2025; Liu et al., 2024b), a phenomenon attributed to the modality gap, where one modality contains more information than the other (Schrodi et al., 2024).

While these studies address general settings, we focus on the specific context of CBMs. This setup presents unique challenges, as concept sets are often highly correlated by design. To our knowledge, the literature lacks a comprehensive evaluation of CLIP's relevance in CBMs, except for the pioneering work by Debole et al. (2025), which assesses the quality of embeddings derived from foundation models. Our study distinguishes itself by addressing concept hallucination in CBMs.

Another underexplored aspect is localization. CBMs implicitly assume that concept representations should not only detect the presence of concepts but also locate them within images. Pre-CLIP CBMs achieved this through backbones trained with localization-aware loss functions (Díaz-Rodríguez et al., 2022; Bennetot et al., 2022). Regarding mitigation methods, Srivastava et al. (2024); Huang & Huang (2024) propose fine-tuning the CLIP backbone to improve localization. The closest work to ours is Yeo et al. (2025), which identifies attention heads responsible for hallucination. However, our method differs in both the identification approach and its application to CBMs. Another noteworthy work is Zhang et al. (2026), which proposes a model-agnostic method based on spatially-aware saliency maps by design.

**Class / concept**   In the context of CBMs, classes refer to the target labels intended for prediction, which are inherently determined by the dataset. Concepts, by contrast, represent a set of interpretable entities—most commonly textual descriptions—that serve as proxies for performing inference. Within CBMs applied to image classification, an image is first represented in terms of these concepts, after which the class prediction is derived from this conceptual representation. In most implementations, concepts correspond to subcomponents of the target label. Two predominant approaches have emerged for defining these concepts. The first one involves prompting large language models: for instance, Yang et al. (2023) extract concepts by querying GPT-3 with prompts such as "describe what the [CLASS NAME] looks like." The second approach leverages dedicated datasets that annotate specific attributes or subparts of the output class present in the image (Díaz-Rodríguez et al., 2022).

## 3.2   Probing CLIP for Concept Hallucination

To study potential limitations of CLIP-based Concept Bottleneck Models (CBMs), we first need to understand what drives a high CLIP score. In this subsection, we design an experiment to test whether CLIP embeddings reliably reflect the physical presence of concepts in images, or whether they are influenced by contextual or semantic cues.

**Datasets**   We perform experiments on three different datasets: ImageNet (Deng et al., 2009), MonumAI (Lamas et al., 2021), and CUB (Wah et al., 2011).

- **ImageNet:** A large-scale object classification dataset where classes correspond to everyday objects (e.g., *kit fox*), and concepts refer to object parts (e.g., *head*, *tail*, *paw*). Since ImageNet lacks fine-grained part annotations, we extended it with PartImageNet++ (Li et al., 2024).

- **MonumAI:** A dataset focused on monument style classification, where classes are architectural styles and concepts correspond to structural elements such as *arches*, *columns*, or *domes*.

- **CUB:** A fine-grained bird classification dataset, where concepts are visual parts such as *wings*, *beak*, or *tail*.

These datasets cover a wide range of tasks, from generic object recognition to fine-grained classification, making them suitable for evaluating concept detection. A complete list of the concepts used in each dataset is provided in Appendix B.

**CLIP Score as a Measure of Concept Detection**   Given an image $I$ and a text $T$, CLIP uses an image encoder $M_{\text{img}}(\cdot)$ and a text encoder $M_{\text{text}}(\cdot)$ to project them into a shared embedding space. The similarity between image and text is computed by the cosine similarity:

$$S(I, T) = \langle M_{\text{img}}(I), M_{\text{text}}(T) \rangle.$$

This score allows CLIP to match images and text without explicit training on the target dataset, which is why it has become a standard tool for zero-shot classification and concept detection. However, if the score is high even when the concept is absent from the image, it indicates a hallucination problem.

**Experimental Setup** We construct three subsets of data given two classes $c_1$ and $c_2$, and a concept $k$ that is strongly linked to $c_1$ but absent from $c_2$:

- Images of class $c_1$ where concept $k$ is present.

- Images of class $c_1$ where concept $k$ is absent.

- Images of class $c_2$, where concept $k$ is always absent by design.

For each triplet $(c_1, c_2, k)$, we sample images randomly and repeat the process 10 times. The full list of triplets is given in the appendix.

The goal of this setup is to test whether CLIP can tell apart the true presence of a concept from its mere semantic association with a class. Concretely, we compute the average CLIP score of each subset with respect to the concept $k$. We also compute the *failure rate*, defined as the fraction of cases where the subset without the concept receives a higher score than the subset where the concept is actually present.

**Results and Analysis** Table 1 reports the average CLIP score across all three datasets. The results show that CLIP does not reliably separate the true presence of a concept from its absence. For example, in both MonumAI and CUB, the scores for $k$-present and $k$-absent subsets of class $c_1$ are almost identical, indicating that CLIP relies heavily on class-level associations rather than visual evidence. The high failure rates (40–50%) further confirm that CLIP often assigns higher scores to images without the concept than to those containing it. This demonstrates a significant risk of *concept hallucination*, raising concerns about using CLIP-based embeddings as faithful representations in CBMs.

|  | $c = c_1$
*k present* | $c = c_1$
*k absent* | $c = c_2$
*(k absent)* | *Fail. Rate* |
|---|---|---|---|---|
| MonumAI | $18.16 \pm 2.45$ | $18.09 \pm 2.38$ | $16.71 \pm 2.75$ | 0.40 |
| CUB | $15.58 \pm 1.74$ | $15.61 \pm 1.65$ | $12.73 \pm 2.06$ | 0.50 |
| ImageNet | $19.35 \pm 1.93$ | $19.18 \pm 1.37$ | $14.82 \pm 1.80$ | 0.40 |

Table 1: **Average CLIP score on different setups.** In the first column, $k$ is present in the images. In the second and third ones, $k$ is not present. *Fail. Rate* presents the failure rate, i.e., the fraction of cases where the subset of images that do not possess the desired concept induces a higher score.

## 4 Disentangling concept representations

### 4.1 Preliminaries

To address this challenge, we introduce a novel method, CHILI, for disentangling concept localization from concept suggestion within the conceptual representation of images. The primary objective of this approach is to provide users with a conceptual representation that decomposes into distinct components: one related to the object of interest and another one to its surrounding context. By selectively retaining only the object-related component, our method aims to produce a conceptual representation that reduces concept hallucinations.

**Notations** We now describe in more detail how the image encoder $M_{\mathrm{img}}$ of CLIP computes its representations. The encoder is a Vision Transformer (ViT) consisting of $L$ stacked transformer layers, each containing a multi-head self-attention (MSA) block and a multi-layer perceptron (MLP) block. The input image $I$ is first split into a sequence of patches, linearly projected into tokens, and augmented with a special *class token* (denoted by index *cls*). These tokens are processed layer by layer through the transformer. Formally, we denote by $h \in [\![1, H]\!]$: the attention heads, $l \in [\![1, L]\!]$: the transformer layers, $i \in [\![1, N]\!]$: the patch tokens, and $Z^l$: the residual stream (intermediate representation) at layer $l$.

The image encoder produces a single vector representation of the image by applying a learned projection $P$ to the final embedding of the class token. In CLIP, this is written as:

$$M_{\text{img}}(I) = P \cdot \left[Z^L\right]_{\text{cls}} ,$$

where $\left[Z^L\right]_{\text{cls}}$ denotes the class token at the final layer.

**Unrolling the Transformer.** Each layer $l$ of the transformer updates the residual stream by combining the outputs of the MSA and MLP blocks:

$$Z^l = Z^{l-1} + \text{MSA}^l\left(Z^{l-1}\right) + \text{MLP}^l\left(\hat{Z}^l\right) ,$$

where $\hat{Z}^l$ denotes the normalized activations after the attention block.

By expanding this recursion, we can express the final class token representation as a sum of contributions from all layers:

$$M_{\text{img}}(I) = P\left[Z^0\right]_{\text{cls}} + \sum_{l=1}^{L} P\left[\text{MSA}^l\left(Z^{l-1}\right)\right]_{\text{cls}} + \sum_{l=1}^{L} P\left[\text{MLP}^l\left(\hat{Z}^l\right)\right]_{\text{cls}} . \tag{1}$$

**Decomposition into Attention Heads.** Following the analysis of Elhage et al. (2021); Gandelsman et al. (2023), the image embedding $M_{\text{img}}(I)$ can be written as the sum of contributions from each transformer layer $l$, each attention head $h$, and each image token $i$. Intuitively, let us consider that a transformer layer has an output linear map (usually denoted $W_O^l$) that mixes the heads and produces the final MSA vector. Applying $W_O^l$ and then the projection $P$ to the `cls` MSA output gives

$$P\left[\text{MSA}^l(Z^{l-1})\right]_{\text{cls}} = P\left(W_O^l\left(\left[\text{Head}_{l,1}; \dots ; \text{Head}_{l,H}\right]\right)\right).$$

For our decomposition it is convenient to view the effect of $W_O^l$ as a linear map applied to each head and then summed. Thus we may write

$$P\left[\text{MSA}^l(Z^{l-1})\right]_{\text{cls}} = \sum_{h=1}^{H} \sum_{i=0}^{N} P\, W_O^{l,h}\left(\alpha_{\text{cls},i}^{l,h}\, v_i^{l,h}\right),$$

where $W_O^{l,h}$ denotes the linear map that extracts the contribution of head $h$ after the usual output projection, $v_i^{l,h} := V^{l,h}\left(Z_i^{l-1}\right)$ is the *value* vector produced for token $i$ by head $h$ at layer $l$, and $\alpha_{\text{cls},i}^{l,h}$ is the attention weight from the class query to token $i$ in head $h$, layer $l$.

We now define the vector contribution coming from token $i$, head $h$, layer $l$ after all linear projections:

$$m_{i,l,h} := P\, W_O^{l,h}\left(\alpha_{\text{cls},i}^{l,h}\, v_i^{l,h}\right). \tag{2}$$

Each $m_{i,l,h}$ is a vector in the same embedding space as $M_{\text{img}}(I)$. With this definition we can rewrite the sum of all MSA contributions compactly:

$$\sum_{l=1}^{L} P\left[\text{MSA}^l(Z^{l-1})\right]_{\text{cls}} = \sum_{l=1}^{L} \sum_{h=1}^{H} \sum_{i=0}^{N} m_{i,l,h}.$$

Using equation 2 and the expansion above, equation 1 becomes

$$M_{\text{img}}(I) = P[Z^0]_{\text{cls}} + \sum_{l=1}^{L} \sum_{h=1}^{H} \sum_{i=0}^{N} m_{i,l,h} + \sum_{l=1}^{L} P\left[\text{MLP}^l(\hat{Z}^l)\right]_{\text{cls}}.$$

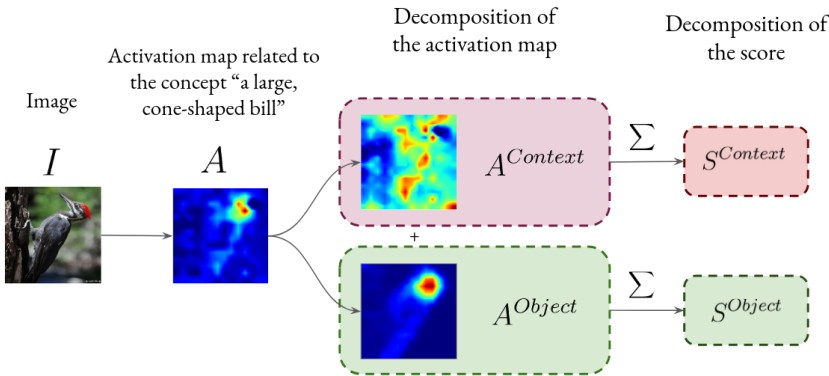

Figure 2: **Decomposition of the activation map**

The first term $P[Z^0]_\text{cls}$ is the projected initial class token; the last sum collects the MLP contributions. Since, by definition, the CLIP score is

$$S(I,T) = \langle M_\text{img}(I), M_\text{text}(T) \rangle,$$

inserting the expression for $M_\text{img}(I)$ gives

$$S(I,T) = \langle P[Z^0]_\text{cls}, M_\text{text}(T) \rangle + \sum_{l=1}^{L}\sum_{h=1}^{H}\sum_{i=0}^{N} \langle m_{i,l,h}, M_\text{text}(T) \rangle + \sum_{l=1}^{L} \langle P[\text{MLP}^l(\hat{Z}^l)]_\text{cls}, M_\text{text}(T) \rangle.$$

Let us define

$$A_{i,l,h}(T) := \langle m_{i,l,h}, M_\text{text}(T) \rangle,$$

which is the scalar alignment between the head/token contribution and the text embedding. If we collect the small terms (initial class-token projection and the MLP outputs) into a residual $\epsilon$, we obtain the compact decomposition proposed by Elhage et al. (2021); Gandelsman et al. (2023):

$$S(I,T) = \sum_{l=1}^{L}\sum_{h=1}^{H}\sum_{i=0}^{N} A_{i,l,h}(T) + \epsilon. \tag{3}$$

Here we have $\epsilon = \langle P[Z^0]_\text{cls}, M_\text{text}(T) \rangle + \sum_{l=1}^{L} \langle P[\text{MLP}^l(\hat{Z}^l)]_\text{cls}, M_\text{text}(T) \rangle$.

Notably, we can represent the contribution of a specific MSA head $h$ and layer $l$ to the score as an attention map by grouping terms $A_{l,i,h}$ by position, which we denote by $A_{l,h} = [A_{l,i,h}]_{i=0}^{N}$. When reshaped, $A_{l,h}$ can represent the heatmap illustrating the patch-wise contributions as a tensor of size $N$. Finally, we denote by $A$ the summed attention map:

$$A = \sum_{l=1}^{L}\sum_{h=1}^{H} A_{l,h}.$$

### 4.2 Concept Hallucination Inhibition via Localized Interpretability (CHILI) − our method

Our goal is to find a decomposition of $A_{l,h}$ into two terms, respectively representing the activations related to the effective presence of the object in the image, and the activations related to the suggestions of the presence of the concept (see Figure 2):

$$A_{l,h} = A_{l,h}^\text{Context} + A_{l,h}^\text{Object}. \tag{4}$$

- $A_{l,h}^\text{Context}$ is linked with all the content in the image that is not the concept, i.e., locations that do not overlap with the segmentation of the concept.

- $A_{l,h}^{\text{Object}}$ is linked with all the content in the image that is the concept, i.e., locations that do overlap with the segmentation of the concept.

**Filtering Pseudo-Register Artifacts**  The first step of our method is to remove *high-norm tokens*, which are known artifacts of Vision Transformers trained on large datasets  (Darcet et al., 2024; Lu et al., 2025).  We refer to these artifacts as *pseudo-registers*[1]: they tend to store global information, but their spatial localization on the activation map is not meaningful for interpretability. Following this intuition, we separate this component and denote it as the *pseudo-register part.*    Formally, for each attention map $A_{l,h}$ at layer $l$ and head $h$, we define the pseudo-register part as the residual after applying a median filter:

$$A_{l,h}^{\text{P. register}} = A_{l,h} - f_m(A_{l,h}),$$

where $f_m$ is a median filter with kernel size 3.  This operation removes localized noise while isolating the global, non-informative artifacts.

**Remaining activations.**  After filtering, we assume that the remaining activations in $A_{l,h}$ represent the *spatially-dependent part* of the conceptual representation. In other words:

- some neurons focus on detecting patterns that directly correspond to the object (concept) described by the text $T$,

- while other neurons detect contextual features that indirectly suggest the presence of $T$.

**Weighting heads and layers.**  To search for such a decomposition, we assign a weight $w_{l,h}$ to each pair $(l, h)$ according to a score based on the Intersection over Union IoU$(\cdot, \cdot)$ between a pseudo mask based on activations and ground truth segmentations obtained using a probing dataset:

$$w_{l,h} = \mathop{\mathbb{E}}_{A_{l,h}, G \in \mathcal{D}_{\text{probe}}} \left[1 - e^{-\alpha \, \text{IoU}(h_m(A_{l,h}), G))}\right],$$

with $\mathcal{D}_{\text{probe}}$ the probing dataset containing activations $A_{l,h}$ and ground truth segmentations $G$ that segment the concept, $\alpha$ is a temperature scaling hyperparameter, and

$$h_m(A_{l,h}) = \begin{cases} 1 & \text{if} \quad f_m(A_{l,h}) > \text{mean}(f_m(A_{l,h})), \\ 0 & \text{if} \quad f_m(A_{l,h}) \leq \text{mean}(f_m(A_{l,h})). \end{cases}$$

In our experiments, the probing dataset corresponds to, for each data point of the training set, the selection of a random concept present in the image, and the corresponding mask. Note that for each test dataset, we considered a different probing set.

**Decomposition of activations.**  Once the decomposition is performed for each head and layer, using these weights, we split each activation map into two parts:

$$\begin{aligned} A_{l,h}^{\text{Object}} &= w_{l,h} \cdot f_m(A_{l,h}), \\ A_{l,h}^{\text{Context}} &= (1 - w_{l,h}) \cdot f_m(A_{l,h}). \end{aligned}$$

Here:

- $A_{l,h}^{\text{Object}}$ captures features directly aligned with the concept (object-related),

- $A_{l,h}^{\text{Context}}$ captures features not aligned with the concept, i.e., contextual cues.

---

[1]Since previous works (Darcet et al., 2024) demonstrate that incorporating registers into the model architecture effectively mitigates these artifacts, we adopt the term "pseudo-registers" for clarity and convenience.  However, we emphasize that this interpretation—while strongly supported by empirical evidence—remains a working hypothesis rather than a definitive conclusion.

|  | *Monumai* | *ImageNet* | *CUB* |
|---|---|---|---|
| LTC (Yeo et al., 2025) | 0.536 (0.020) | 0.566 (0.016) | 0.532 (0.013) |
| Concept Attention (Gandelsman et al., 2023) | 0.516 (0.018) | 0.495 (0.022) | 0.485 (0.014) |
| Register | 0.548 (0.018) | 0.495 (0.039) | 0.487 (0.019) |
| CHILI (ours) | **0.565 (0.015)** | **0.596 (0.017)** | **0.533 (0.020)** |

Table 2: **Performance comparison across datasets on concept detection.** Results are shown for different methods (LTC, CHILI, Register, and Concept Attention) on three datasets: Monumai, ImageNet, and CUB. Values represent the mean AUC with variance averaged over different runs.

**Score decomposition.** By combining equation 5, equation 4, and the resummations among tokens:

$$S^{\text{Object}} = \sum_{i=0}^{N} A_i^{\text{Object}}$$

$$S^{\text{Context}} = \sum_{i=0}^{N} A_i^{\text{Context}},$$

we can also disentangle the CLIP score into two interpretable contributions:

$$S(I,T) = S^{\text{Object}} + S^{\text{Context}} + \epsilon. \tag{5}$$

The activations $A^{\text{Object}} = \sum_{l=1}^{L} \sum_{h=1}^{H} A_{l,h}^{\text{Object}}$ and $A^{\text{Context}} = \sum_{l=1}^{L} \sum_{h=1}^{H} A_{l,h}^{\text{Context}}$ can thus be interpreted as the token-level contributions to the scores $S^{\text{Object}}$ and $S^{\text{Context}}$, respectively, for the given image and text.

For further details, we invite the reader to look at Appendix C that provides observations about the pseudo-register filtering method, the isolated contribution of each component, the temperature parameter, and the probing dataset.

## 5 Experiments

### 5.1 Concept detection

The most straightforward way to evaluate the efficiency of our method is to evaluate it on a binary object detection task. To do so, we use the same setup as the statistical analysis in Section 3.2. From this setup, we compute the AUROC score in the case where the class present in the image is intended to (column 1 vs column 2 of Table 1) using the different components $S$ (refered as Concept Attention)$S^{\text{Object}}$ (refered as CHILI) $S^{\text{Object}}$ (refered as CHILI in the table), $S^{\text{Context}}$, $S^{\text{P.register}} = \sum_{i=0}^{N} \sum_{l=1}^{L} \sum_{h=1}^{H} A_{i,l,h}^{\text{P.register}}$ from the decomposition of equation 5, and locate-then-correct (LTC) (Yeo et al., 2025). The results are displayed in Table 2.

First, we observe that the baseline—which corresponds to using the raw CLIP score $S$—struggles to detect the presence of the concept, thereby reinforcing the findings of Section 3. Regarding the disentangled components, the $S^{\text{Context}}$ component also exhibits poor detection performance. In contrast, the $S^{\text{Object}}$ component demonstrates a significantly higher detection capability, as intended by design.

The use of the pseudo register component $S^{\text{P.register}}$ showcases similar performances to using the raw CLIP score, indicating that pseudo registers contain non-localised, hallucination-prone information.

### 5.2 Object segmentation

To test whether our method can localize concepts in images, we adapt it into a segmentation module. The task is to segment both *classes* and *concepts* from ImageNet across the entire test set. In practice, we evaluate how well the activation maps highlight the relevant pixels using three standard metrics:

- **Pixel accuracy (Acc.)** — percentage of correctly classified pixels,

- **mean Intersection over Union (mIoU)** — overlap between prediction and ground truth masks,

- **mean Average Precision (mAP)** — quality of the predicted mask in terms of precision.

We compare our method with several post-hoc interpretability approaches (i.e., without fine-tuning the model): LRP (Binder et al., 2016), partial-LRP (Voita et al., 2019), rollout (Abnar & Zuidema, 2020), raw attention, Grad-CAM (Selvaraju et al., 2017), Chefer et al. (Chefer et al., 2021), and Concept Attention (Gandelsman et al., 2023). Among them, two are natural baselines for the *concept-level segmentation*:

- **Raw attention:** the penultimate layer of the vision transformer ($M_{\mathrm{img}}(I)$),

- **Concept Attention:** the original, non-disentangled activation map $A$.

Table 3 reports the results. We highlight the scores of our *Object* component (CHILI) $A^{\mathrm{Object}}$.

| Method | Pixel Acc. ↑ | mIoU ↑ | mAP ↑ |
|---|---|---|---|
| *Class-level segmentation* | | | |
| LRP | 52.81 | 33.57 | 54.37 |
| partial-LRP | 61.49 | 40.71 | 72.29 |
| rollout | 60.63 | 40.64 | 74.47 |
| raw attention | 65.67 | 43.83 | 76.05 |
| GradCAM | 70.27 | 44.50 | 70.30 |
| Chefer et al. | 69.21 | 47.47 | 78.29 |
| Concept Attention | 76.78 | 57.14 | 82.89 |
| CHILI (ours) | **78.79** | **60.22** | **84.86** |
| *Concept-level segmentation* | | | |
| Concept Attention | 70.86 | 46.17 | 87.65 |
| CHILI (ours) | **71.76** | **47.74** | **88.38** |

Table 3: **Segmentation performance on ImageNet.** Results for class-level segmentation (top) and concept-level segmentation (bottom). Higher is better.

**Analysis.** From the class-level results, we observe that our method outperforms all baselines across all three metrics. In particular:

- Compared to **Concept Attention**, our disentangled *Object* map improves pixel accuracy by +2.0 points ($76.78 \rightarrow 78.79$), mIoU by +3.1 points ($57.14 \rightarrow 60.22$), and mAP by +2.0 points ($82.89 \rightarrow 84.86$).

- The improvement over other classical methods such as Grad-CAM (+8.5 mIoU) or rollout (+19.6 mAP) is even more pronounced.

At the concept level, we also see consistent but smaller gains: about +1 point in accuracy, +1.5 in mIoU, and +0.7 in mAP. These results confirm that isolating the *Object* component leads to cleaner and more accurate localization than using the full, entangled activation map. This demonstrates the interest of CHILI.

**Remark.** From an XAI perspective, simply providing accurate object segmentations is not enough to build a reliable Concept Bottleneck Model (CBM). In the next section, we show how CHILI can be leveraged to construct a CBM that is not only accurate but also trustworthy.

# 6 Applying CHILI to CBMs

## 6.1 Method

In the previous sections, we showed how CHILI allows us to mitigate the localisation issue of CBMs, leading to more interpretable concept extraction. We now turn to the question of how this disentanglement impacts the performance of CBMs.

**Baseline.** As a baseline, we consider a standard CLIP-based CBM (Yan et al., 2023b), which relies directly on the full CLIP similarity score $S(I, T)$. In contrast, our approach replaces this score with the disentangled *object-only* component $S^{\text{Object}}$, introduced in Section 4.

Since $S^{\text{Object}}$ is less affected by contextual bias and concept hallucination, we hypothesize that it can yield a more interpretable CBM, albeit at the possible cost of predictive accuracy.

**Evaluation.** For each dataset, we compare the classification accuracy of the baseline CBM (using $S$) with the proposed CBM (using $S^{\text{Object}}$). The results are reported in Table 4.

| Method | *Monumai* | *ImageNet* | *CUB* |
|---|---|---|---|
| Baseline CBM ($S$) | 74.67 | 73.55 | 65.05 |
| CHILI (ours, $S^{\text{Object}}$) | 74.34 | 72.80 | 64.90 |

Table 4: **Classification accuracy of CBMs with and without CHILI.** Results are shown for three datasets. Baseline CBM uses the full similarity score $S$, while our approach uses the disentangled score $S^{\text{Object}}$.

**Analysis.** Our results show that applying CHILI leads to only a minor decrease in accuracy across datasets, suggesting that it can be a viable strategy for improving CBM interpretability without severely compromising performance. The effect depends on the dataset:

- **CUB:** There is almost no accuracy loss, which suggests that object-related signals dominate the decision process in this dataset.

- **Monumai and ImageNet:** A small drop in accuracy occurs, likely because contextual features (e.g., background or environment) play a role in classification. By removing them, the model becomes less biased but also loses some useful cues.

**Discussion.** Overall, these findings highlight a trade-off:

- On the positive side, CHILI reduces bias and mitigates errors caused by spurious correlations, especially in fine-grained datasets such as CUB.

- On the negative side, filtering out contextual information inevitably discards some predictive features, which can slightly reduce accuracy.

Importantly, there is no guarantee that reducing hallucinations will improve accuracy. However, from an XAI perspective, prioritizing faithfulness and interpretability is crucial, and our results suggest that CHILI can achieve this while keeping performance reasonably close to the baseline.

## 6.2 Explanations

Once the model is trained, we can leverage the activations produced to build visual explanations that gather the name of the most important concept and their location on the image. We explain the process below.

We extract the concept representation using CHILI. We then apply DeepSHAP (Lundberg & Lee, 2017) to the model, with a key distinction: rather than computing SHAP values at the image level, as is conventional

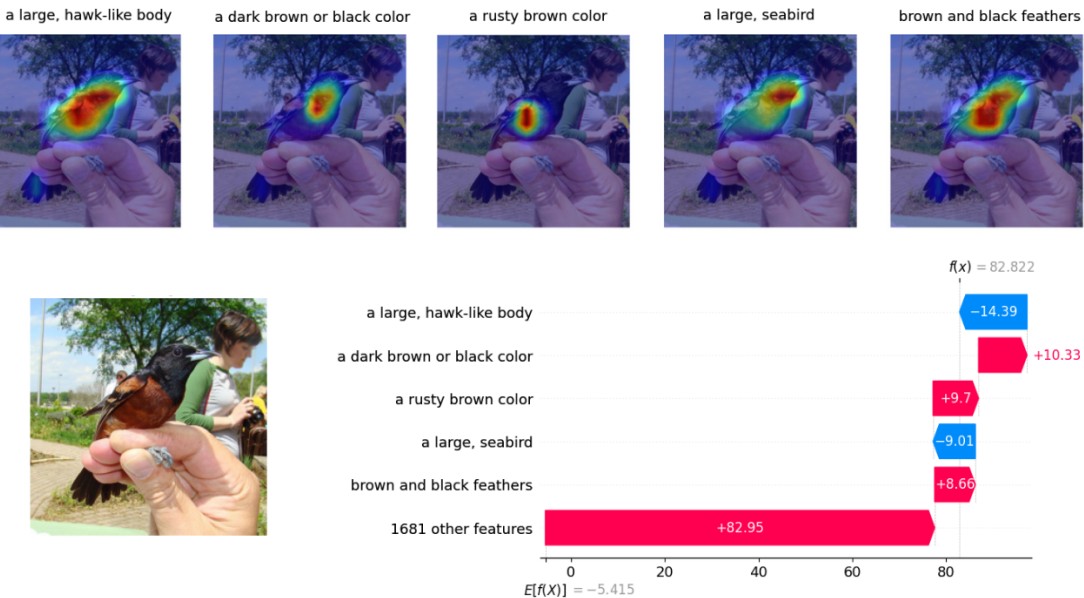

Figure 3: **Example of explanation produced by the intervention of CHILI in a CBM.** On the bottom left, the input image. On the bottom right, the SHAP values. Target label: *Orchard Oriole*

for image classification tasks, we perform the computation at the concept level. This approach allows us to quantify the importance of each concept in the inference of the target label. Finally, for the top five concepts identified by DeepSHAP, we visualize their contributions as heatmaps derived from their corresponding activations, $A^{\text{Object}}$. Examples of these explanations, including failures cases, are presented in Figure 3 and Appendix E.

## 7 Limitations and discussion

In this work, we studied the ability of CLIP to focus on patterns located in the object designated by the textual encoding to produce an inference and proposed a way to disentangle the activations of the model without fine-tuning. We want to discuss here the limitations of such a procedure.

**Layer/head decomposition**   We base our approach on a layer/head decomposition to achieve the disentangling. Such an assumption, as noted by Gandelsman et al. (2023), neglects indirect effects, i.e., potential interactions from previous layers on deeper ones. Additionally, we assume that each position in the layer/head pairs can be attributed to pseudo-register, context, or object (or at least can be more easily separated by doing so).

**Other factors of a high CLIP score**   We voluntarily focus on the impact of the object's presence on the increase or decrease of the CLIP score. However, being a complex model, the factors that influence high CLIP scores are multifactorial. For example, the proportion of salient features (Darcet et al., 2024) or the text prompt (Zhou et al., 2022) also influences its output. Such factors are a notable reason why our disentangling does not completely eliminate the failures in the experiments of Section 3.2. For example, the images from the case $c = c_1; k \, absent$ have by construction many more close-ups than the case $c = c_1; k \, present$, inducing perturbation towards higher scores. It is also evident that CLIP suffers from numerous biases (Moayeri et al., 2023b) that can influence the score in either a decreasing or increasing manner.

**Role of pseudo registers**   One aspect of our disentangling is the presence of high-norm artifacts (Darcet et al., 2024), which we refer to as pseudo-registers. The reason we dedicated a special part to them in our

decomposition is that their role is ambiguous: since they do not seem to exhibit spatial coherence with the image, it is difficult to determine whether they store information about the object or the concept.

## 8 Conclusion

In this paper, we examined the limitations of using CLIP as a concept extractor. Through statistical analysis, we identified challenges associated with correlating high scores with the localization of concepts in images, particularly in cases where the presence of a concept is merely suggested. To address this issue, we propose a method that factorizes the embedding space into components related to the object, the context, and pseudo-registers. Empirical results demonstrate that our disentangling approach can partially eliminate the contextual aspects of conceptual representation thereby advancing towards more localization-focused CLIP-based concept bottleneck models. In addition, while a probing dataset is required to compute the calibration performed in our method, our approach does not require any additional training of the model.

However, several limitations are present in our study. Primarily, we were unable to achieve complete disentanglement of the activations. This shortcoming can be attributed to multiple hypotheses we adopted in our experimental setup. Notably, we neglected second-order effects and assumed that attention heads are not polysemantic, an assumption that is somewhat reductive.

In this paper, we limit ourselves to CLIP based on ViT backbones. This restriction is motivated by the fact that this paradigm has become the gold standard for most zero-shot CBMs released in recent years (Yang et al., 2023; Yan et al., 2023a; Kazmierczak et al., 2024). However, we plan to extend these analyses to other overlooked backbones, such as those based on ResNet, and models, such as LLaVa. The goal is twofold: first, this opens the way to post-hoc disentangling on other CBMs; secondly, it allows us to study the similarities and differences across various embedding networks, potentially leading to more interpretable zero-shot CBMs.

## 9 Acknowledgments

This work was performed using HPC resources from GENCI-IDRIS (Grant 2025 - AD011014675R2)

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

## A    Use of CLIP in CBMs

Table 5 presents the foundation models used in the CBMs highlighted in the study of Kazmierczak et al. (2025). Note that the list is not exhaustive.

## B    Datasets

**ImageNet**    The first dataset we use is ImageNet (Deng et al., 2009), that provides annotation of images into 1000 classes. The dataset having not concept-level annotations natively, we used the PartImageNet++ dataset (Li et al., 2024), which provides semantic segmentation annotations for different images of ImageNet. The scenarios used in our study are detailed in Table 6.

| Title | PFM used |
|---|---|
| STAIR (Chen et al., 2023) | CLIP |
| Chat GPT XAI (Liu et al., 2023) | CLIP |
| ARTxAI (Fumanal-Idocin et al., 2023) | CLIP |
| Explainable meme classification (Thakur et al., 2022) | CLIP |
| Label free CBM (Oikarinen et al., 2023) | CLIP |
| LaBo (Yang et al., 2023) | CLIP |
| Learning Concise (Yan et al., 2023a) | CLIP |
| Sparse CBM (Panousis et al., 2023) | CLIP |
| CBM with filtering (Kim et al., 2023) | CLIP |
| Robust CBM (Yan et al., 2023b) | CLIP |
| Hierarchichal CBM (Panousis et al., 2024) | CLIP |
| ChatGPT CBM (Ren et al., 2023) | CLIP |
| Skin lesion CBM (Patrício et al., 2024) | CLIP |
| R-VLM (Xu et al., 2023) | CLIP |
| CEIR (Cui et al., 2023) | CLIP |
| Concept Gridlock (Echterhoff et al., 2024) | CLIP |
| SpLiCE (Bhalla et al., 2024) | CLIP |
| MMCBM (Wu et al., 2025) | CLIP |
| XCoOp (Bie et al., 2024) | CLIP |
| CLIP-QDA (Kazmierczak et al., 2024) | CLIP |
| Text-To-Concept (Moayeri et al., 2023a) | CLIP |
| LLM-Mutate (Chiquier et al., 2024) | Llama2+CLIP |
| VAMOS (Wang et al., 2024) | BLIP-2 |
| Q-SENN (Norrenbrock et al., 2024) | CLIP |
| ExTraCT (Yow et al., 2024) | CLIP+BERT |
| Adaptative CBM (Choi et al., 2024) | CLIP |
| Stochastic CBM (Vandenhirtz et al., 2024) | CLIP |
| Med-MICN (Hu et al., 2024) | CLIP |
| VLG-CBM (Srivastava et al., 2024) | CLIP |

Table 5: **PFM usage in CBMs.**

| Scenario | $c_1$ | $c_2$ | $k$ |
|---|---|---|---|
| 1 | Tiger_cat | Gondola | tail |
| 2 | Bolete | Stole | lamellae |
| 3 | LesserPanda | BlackSwan | paw |
| 4 | ModelT | Turnstile | wheel |
| 5 | Plunger | CommonIguana | handle |
| 6 | AnalogClock | Goldfish | dial |
| 7 | Fly | Strawberry | wing |
| 8 | Barracouta | Barbell | fin |
| 9 | ComputerKeyboard | Convertible | key |
| 10 | FountainPen | HowlerMonkey | ink_cartridge |

Table 6: **List of ImageNet runs with respective triplets classes $c_1$, $c_2$, and concepts $k$.**

**Monumai** Monumai (Lamas et al., 2021) is a specialized dataset containing images of monuments. It is composed of 908 images. Each image is annotated accordingly to the overall structure that corresponds to the class, and the architectural features that corresponds to the concepts. There are 15 concepts and 4 classes available. The scenarios used in our study are detailed in Table 7.

| Scenario | $c_1$ | $c_2$ | $k$ |
|----------|-------|-------|-----|
| 1 | hispanic-muslim | baroque | lobed_arch |
| 2 | baroque | renaissance | porthole |
| 3 | baroque | gothic | broken_pediment |
| 4 | baroque | renaissance | solomonic_column |
| 5 | gothic | hispanic-muslim | pointed_arch |
| 6 | renaissance | baroque | serliana |
| 7 | gothic | baroque | trefoil_arch |
| 8 | baroque | renaissance | rounded_arch |
| 9 | gothic | renaissance | ogee_arch |

Table 7: **List of Monumai runs with respective triplets classes $c_1$, $c_2$, and concepts $k$.**

**CUB**  CUB (Wah et al., 2011), is a dataset dedicated to the classification of birds, with 200 classes corresponding to species. Concept level, localised annotations are also not available natively. To obtained such annotation, we used the procedure of VLG-CBM (Srivastava et al., 2024) that uses GroundingDino (Liu et al., 2024a) to localize concepts as bounding boxes. The scenarios used in our study are detailed in Table 8.

| Scenario | $c_1$ | $c_2$ | $k$ |
|----------|-------|-------|-----|
| 1 | Orchard Oriole | Least Auklet | long tail |
| 2 | Red headed Woodpecker | Bay breasted Warbler | long pointed beak |
| 3 | Worm eating Warbler | Chuck will Widow | yellowish belly |
| 4 | Whip poor Will | Rock Wren | brown or grayish body |
| 5 | House Sparrow | Belted Kingfisher | brown streaks on the chest |
| 6 | Herring Gull | Worm eating Warbler | black wingtips |
| 7 | Ring billed Gull | Red breasted Merganser | white body with gray wings |
| 8 | Red bellied Woodpecker | Red breasted Merganser | white front |
| 9 | Golden winged Warbler | Geococcyx | white belly |
| 10 | Pied Kingfisher | Vermilion Flycatcher | black back |

Table 8: **List of CUB runs with respective triplets classes $c_1$, $c_2$, and concepts $k$.**

## C  Ablation studies

### C.1  Comparative performance analysis

Table 9 presents results for CHILI, presented in the format of Table 1, are reported below. To ensure comparability, all values have been normalized

| Dataset | Base | c = c1; k in c1 (present) | c = c1; k not in c1 (absent) | Failure rate |
|---------|------|---------------------------|------------------------------|--------------|
| ImageNet | Baseline | 0.853 (0.005) | 0.846 (0.005) | 0.4 |
|          | Chili | 0.713 (0.035) | 0.668 (0.040) | 0.2 |
| MonumAI | Baseline | 0.828 (0.025) | 0.768 (0.021) | 0.4 |
|          | Chili | 0.680 (0.035) | 0.712 (0.039) | 0.3 |
| CUB | Baseline | 0.779 (0.029) | 0.794 (0.030) | 0.5 |
|     | Chili | 0.507 (0.070) | 0.548 (0.082) | 0.3 |

Table 9: **Score comparison between original CLIP-scores and CHILI** Normalized values. Variances in parentheses.

The values in parentheses represent standard variances. Consistent with the findings in Section 5, our results indicate that CHILI yields a modest yet measurable improvement, reflected in both the relative score reduction and failure rate decrease.

## C.2 Artifact Mitigation Methods

To address pseudo-register artifacts, we systematically evaluated multiple approaches: erosion/dilation, Principal Component Analysis (PCA), thresholding, and median filtering. Our decision to adopt median filtering was informed by an ablation study, detailed in Table 10, assessing their impact on object segmentation performance.

| Method | pixAcc | mIoU | mAP |
|---|---|---|---|
| Baseline | 0.7678 | 0.5714 | 0.8289 |
| + Erosion/Dilation | 0.7474 | 0.4795 | 0.7070 |
| + PCA | 0.6627 | 0.4503 | 0.7525 |
| + Thresholding | 0.7700 | 0.5775 | 0.8295 |
| + Median Filtering (Ours) | **0.7827** | **0.5945** | **0.8460** |

Table 10: **Performance comparison of pseudo-register artifacts mitigation methods**

Median filtering and thresholding emerged as the most effective methods, likely due to the high-norm, localized nature of the artifacts. We ultimately selected median filtering for its superior performance and to avoid introducing an additional hyperparameter (as required by thresholding).

## C.3 Individual Components

We propose here to measure the apport of pseudo-register filtering of the weighting scheme by isolating them individually. Table 11 echoes the segmentation experiment of Table 3 while Table 12 echoes the segmentation experiment of Table 2.

| Filter Register | Weighting | pixAcc | mIoU | mAP |
|---|---|---|---|---|
| FALSE | FALSE | 0.7678 | 0.5714 | 0.8289 |
| TRUE | FALSE | 0.7827 | 0.5945 | 0.8460 |
| FALSE | TRUE | 0.7735 | 0.5827 | 0.8333 |
| TRUE | TRUE | **0.7879** | **0.6022** | **0.8486** |

Table 11: **ImageNet class segmentation performance comparison with different configurations**

| Filter Register | Weighting | AUROC |
|---|---|---|
| FALSE | FALSE | 0.495 |
| TRUE | FALSE | 0.566 |
| FALSE | TRUE | 0.570 |
| TRUE | TRUE | **0.596** |

Table 12: **ImageNet binary classification performance comparison with different configurations**

In both cases, each component individually improves performance, with the combined approach yielding the best results.

## C.4 Temperature parameter $\alpha$

We analyse here the sensitivity of our method to the hyperparameter $\alpha$, which controls the weighting scheme, in Table 13.

| $\alpha$ | AUROC |
|---|---|
| 1 | 0.586 |
| 5 | 0.582 |
| 10 | **0.596** |
| 50 | 0.587 |
| 100 | 0.587 |

Table 13: **AUROC values for different Temperature parameter $\alpha$.**

The optimal value of $\alpha = 10$ reflects a trade-off: as $\alpha$ approaches 0, weights become insensitive to IoU, while as $\alpha$ approaches infinity, weights approximate an indicator function ($IoU > 0$).

### C.5   Probing dataset

In our experiments, the probing dataset is distinct for each dataset. To further explore the robustness of our approach, we investigated the impact of applying the weights derived from the ImageNet dataset to perform disentanglement on images from another dataset (MonuMAI) and vice versa. The results of this cross-dataset evaluation are presented in the table below.

| Probing dataset | Test dataset | Auroc |
|---|---|---|
| Imagnet | Monumai | 0.557 (0.015) |
| Monumai | Monumai | 0.565 (0.015) |
| Monumai | Imagnet | 0.589 (0.016) |
| Imagnet | Imagnet | 0.596 (0.017) |

Table 14: Auroc values for different probing and test dataset combinations.

While we observe an expected decrease in performance—attributable to the fact that the probing data originates from a different distribution than the inference set—this decline is modest. We hypothesize that this robustness stems from similarities in image content across datasets. This suggests that the learned representations retain a degree of generalizability despite differences in annotation granularity.

## D   Computational costs

All the experiments were conducted on a single V100 GPU with 32GB of RAM. While CHILI introduces additional computational overhead, the memory cost is negligible since the only additional information is the weight matrix. The architecture remains the same. Concerning inference time, as an illustration, we currently average the following times:

| Method | Iters / second |
|---|---|
| Baseline | 51.64 |
| CHILI | 1.13 |

Table 15: Inference time comparison in iterations per second.

While there is a significant increase of inference time, we consider it as reasonable since the architecture allows to precompute CLIP scores in order to train the CBM.

# E    Additional samples

In addition of the explanation samples provided in the main paper, we provide here additional samples. The goal is to provide a more global insight, including failures cases of CHILI. These examples correspond to Figures 4 to 12.

A recurrent issue is the tendency of heatmaps to overfocus on specific regions of the image—such as the heads of birds in Figures 4 or 8—resulting in the mislocalization of certain features. We hypothesize that this behavior stems from the use of a uniform weighting scheme across all concepts. We also observed that our method faces challenges with shape-related concepts, like the concept *a large, hawk-like body* on Figure 9 or *long, thin antenae* in Figure 10.

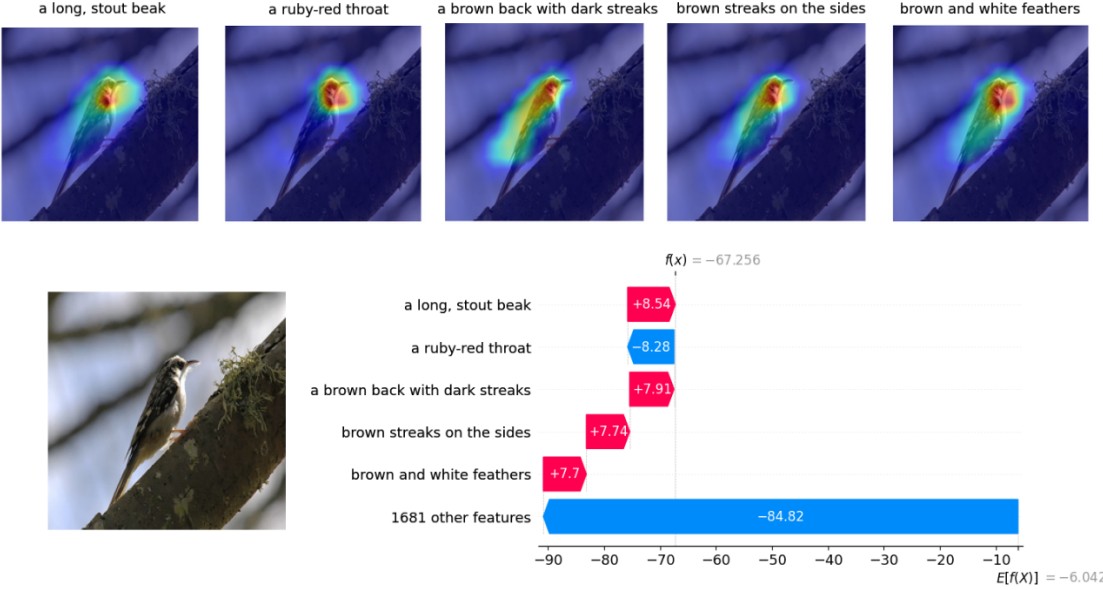

Figure 4: **Example of explanation produced by the intervention of CHILI in a CBM.** On the bottom left, the input image. On the bottom right, the SHAP values. Target label: *Brown Creeper.*

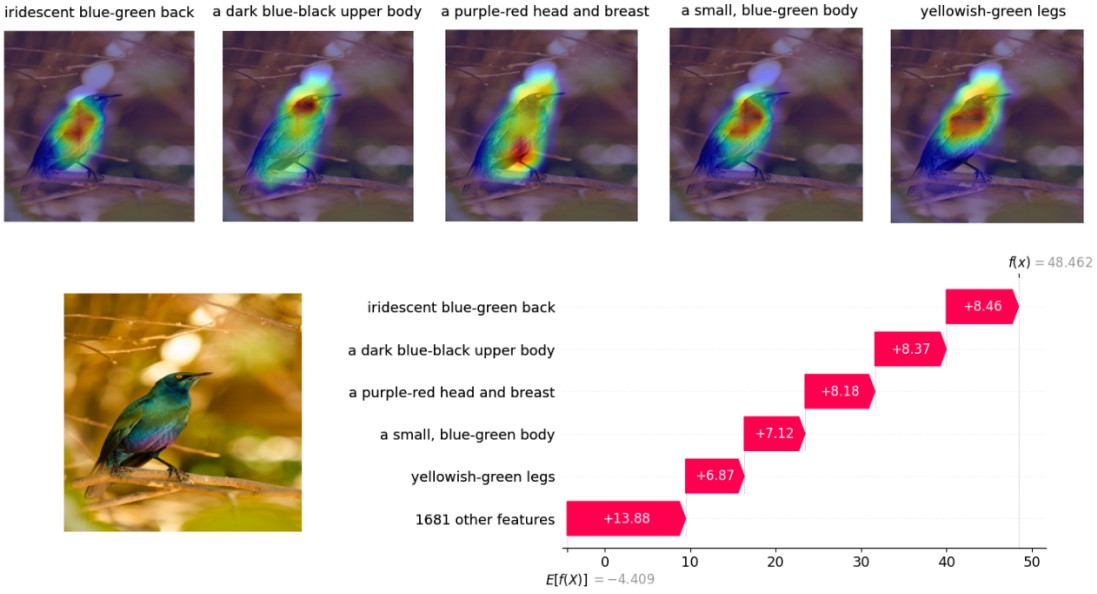

Figure 5: **Example of explanation produced by the intervention of CHILI in a CBM.** On the bottom left, the input image. On the bottom right, the SHAP values. Target label: *Cape Glossy Starling.*

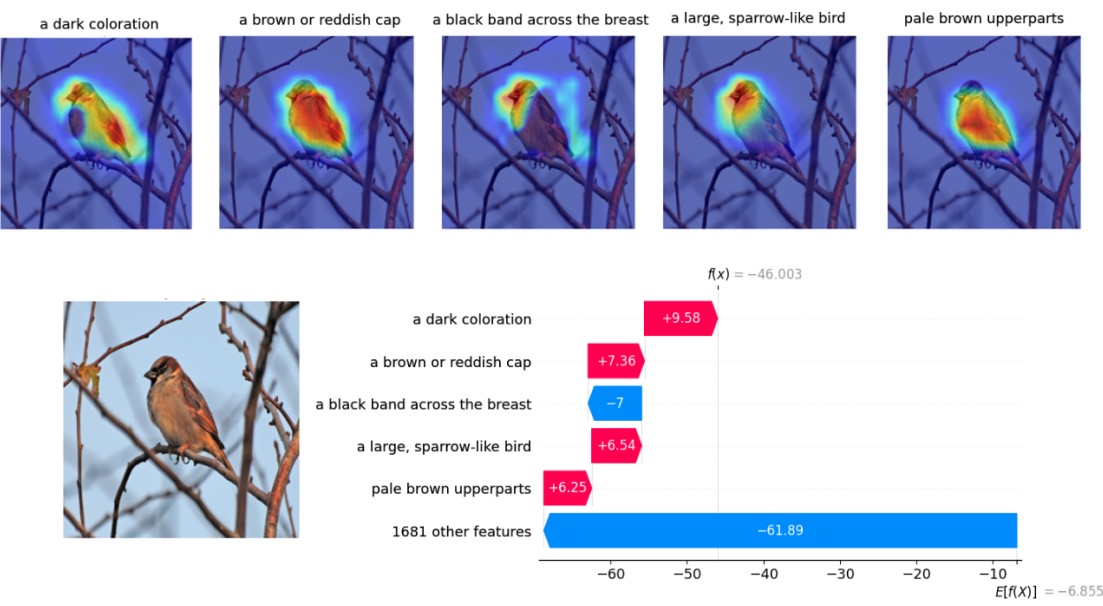

Figure 6: **Example of explanation produced by the intervention of CHILI in a CBM.** On the bottom left, the input image. On the bottom right, the SHAP values. Target label: *House Sparrow.*

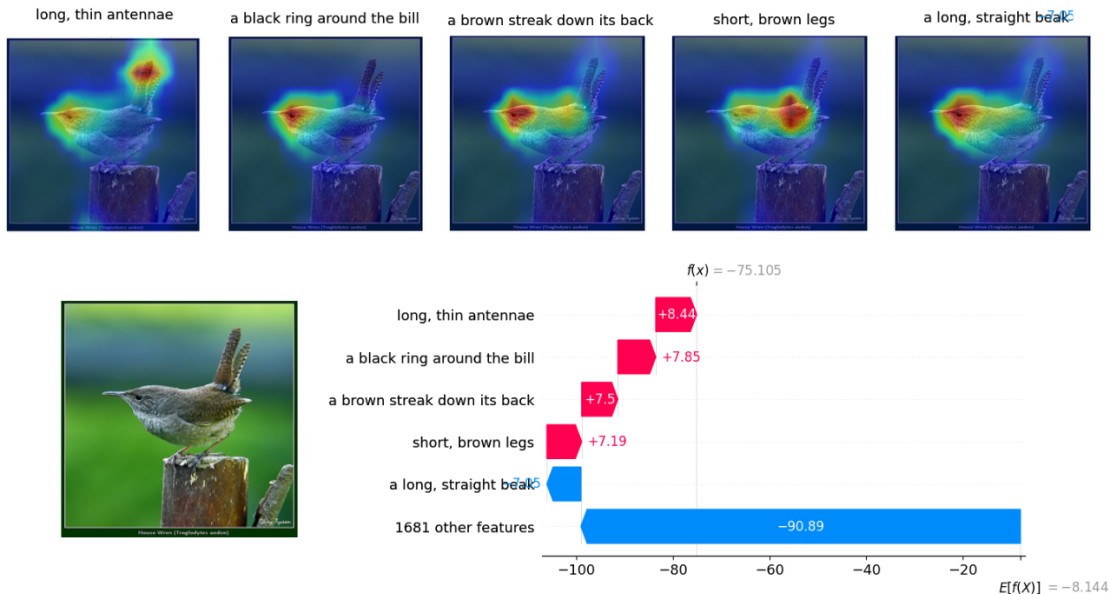

Figure 7: **Example of explanation produced by the intervention of CHILI in a CBM.** On the bottom left, the input image. On the bottom right, the SHAP values. Target label: *House Wren.*

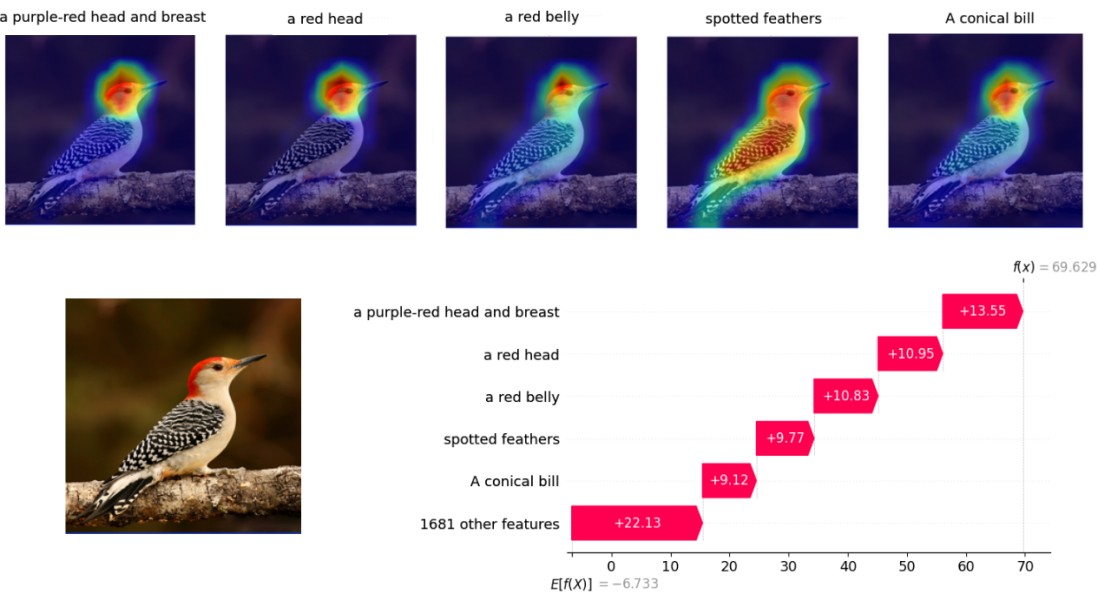

Figure 8: **Example of explanation produced by the intervention of CHILI in a CBM.** On the bottom left, the input image. On the bottom right, the SHAP values. Target label: *Red Bellied Woodpecker.*

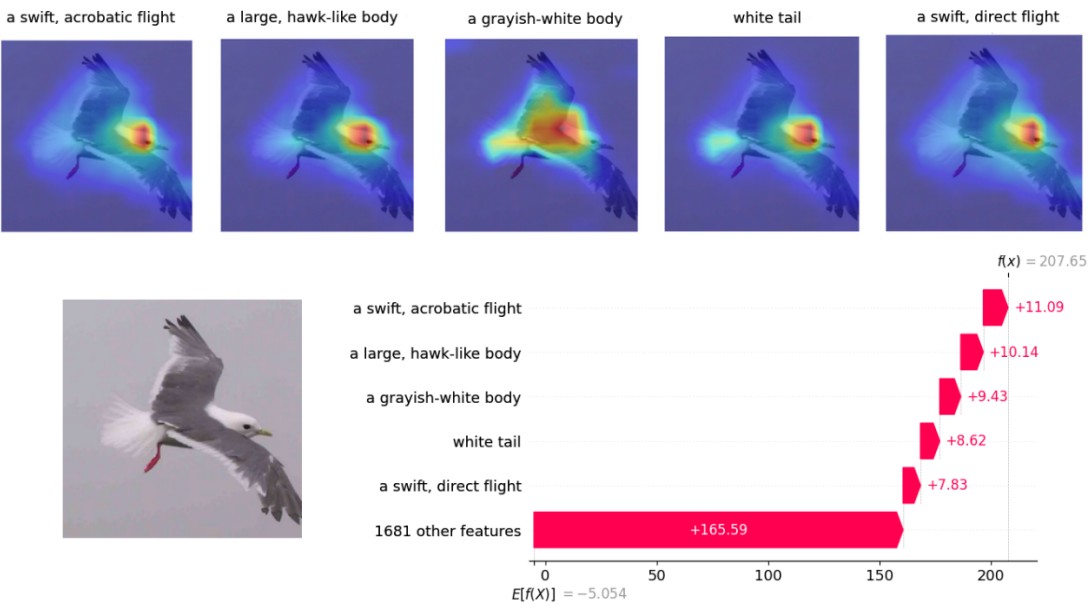

Figure 9: **Example of explanation produced by the intervention of CHILI in a CBM.** On the bottom left, the input image. On the bottom right, the SHAP values. Target label: *Red Legged Kittiwake.*

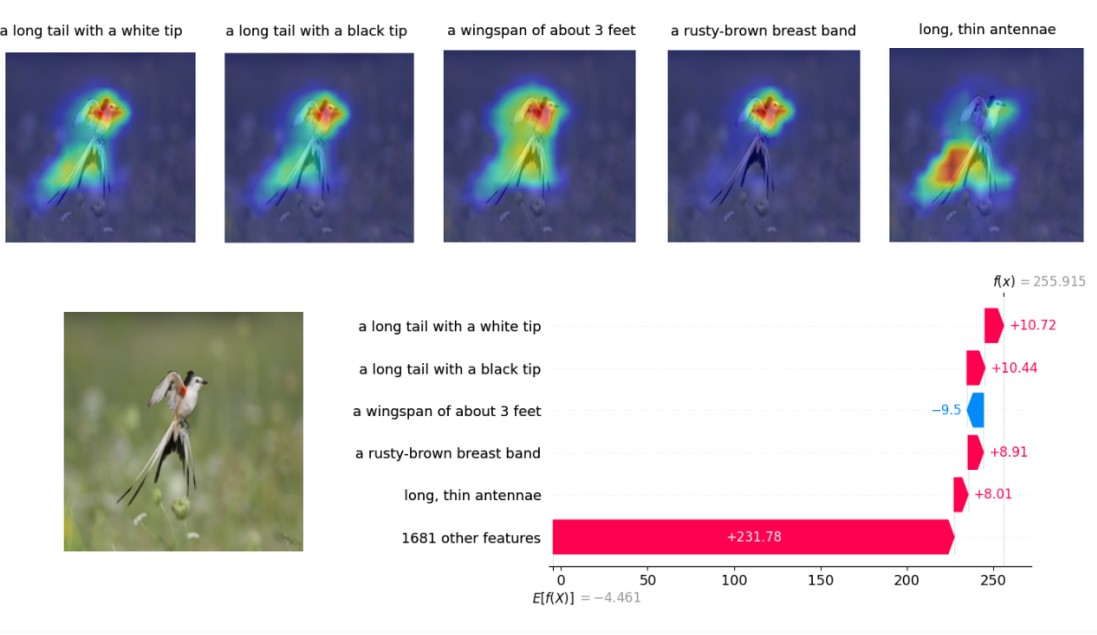

Figure 10: **Example of explanation produced by the intervention of CHILI in a CBM.** On the bottom left, the input image. On the bottom right, the SHAP values. Target label: *Scissor Tailed Flycatcher.*

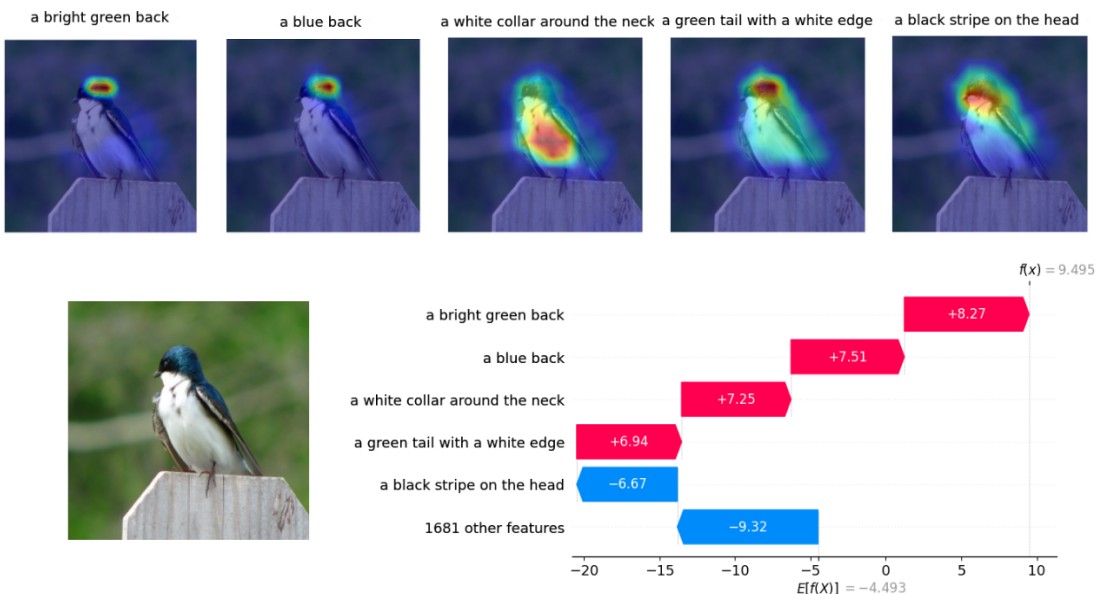

Figure 11: **Example of explanation produced by the intervention of CHILI in a CBM.** On the bottom left, the input image. On the bottom right, the SHAP values. Target label: *Tree Swallow.*

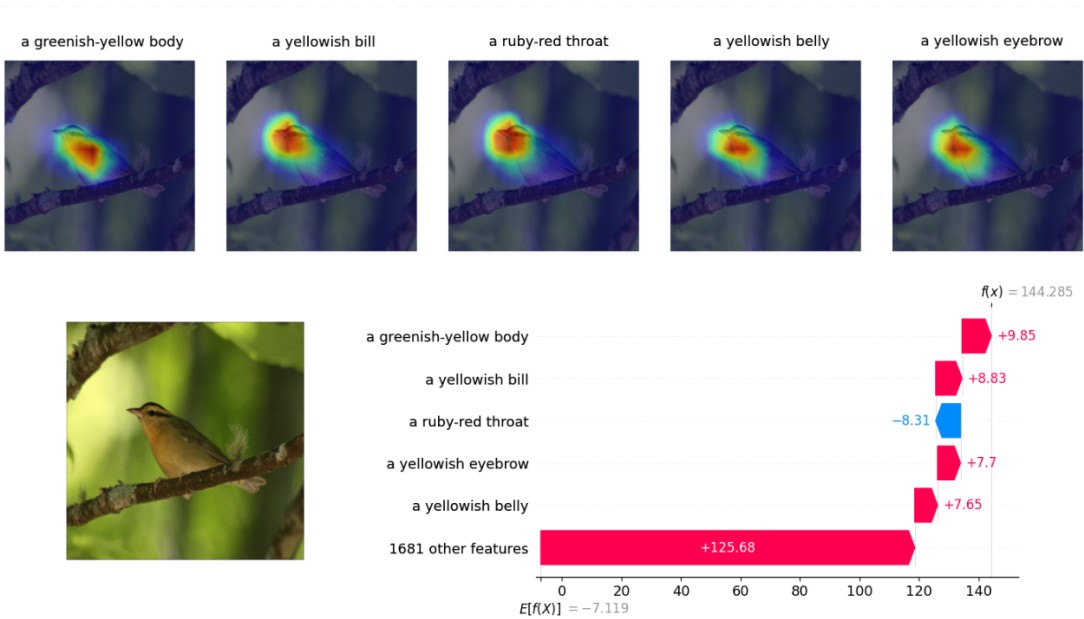

Figure 12: **Example of explanation produced by the intervention of CHILI in a CBM.** On the bottom left, the input image. On the bottom right, the SHAP values. Target label: *Worm Eating Warbler.*

