# OpenReview forum: "Enhancing Concept Localization in CLIP-based Concept Bottleneck Models"
_TMLR — Accepted by TMLR_

### Review · Reviewer_LVKt · 2025-10-23

**Summary Of Contributions:**

This paper presents CHILI (Concept Hallucination Inhibition via Localized Interpretability), a novel, post-hoc method designed to improve the faithfulness and localization of concepts within Concept Bottleneck Models (CBMs) that utilize CLIP for zero-shot concept extraction .
The core contribution is demonstrating and mitigating concept hallucination in CLIP , a phenomenon where high similarity scores reflect contextual or semantic cues instead of the concept's actual physical presence. CHILI addresses this by decomposing the CLIP embedding and score into three distinct, interpretable components: the Object component (linked to physical location), the Context component (linked to features suggesting presence but not directly representing the concept), and a Pseudo-Register component (high-norm artifacts) . This decomposition is achieved by calibrating the contribution of each attention head and layer using an IoU-based metric derived from a dedicated probing dataset containing ground truth segmentations . The resulting Object score then serves as the faithful, localized conceptual representation in the CBM.

**Audience:**

Yes

**Audience Explanation:**

The paper addresses critical issues highly relevant to the TMLR audience, spanning XAI, interpretability, and Vision-Language Models (VLMs). The mitigation of concept hallucination enhances the trustworthiness of CBMs, a key model family for interpretable AI. The analysis of CLIP's internal attention mechanisms and the development of a post-hoc, IoU-calibrated decomposition method provides a valuable analytical tool for VLM researchers. The findings are directly applicable to building more faithful AI systems in sensitive domains.

**Broader Impact Concerns:**

No Impact Concerns.

**Claims And Evidence:**

Yes

**Claims Explanation:**

The claims are supported by strong, quantitative evidence. First, the statistical test in Section 3.2 clearly justifies the problem by showing that the raw CLIP score fails to separate concept presence from semantic association, with up to 50% failure rates in tested scenarios. Second, the efficacy of the proposed disentanglement is proven by the superior performance of the Object component in binary concept detection (Table 2), where it significantly outperforms the baseline CLIP score and concurrent methods. Third, the localization capability is confirmed by the segmentation results (Table 3), with the Object activation map improving mIoU by +3.1 points over the raw attention baseline. Finally, the trade-off is validated by showing that applying CHILI to CBMs results in only a minor, acceptable loss in predictive accuracy.

**Requested Changes:**

The submission would be significantly strengthened by addressing a key area regarding the calibration mechanism and evidence. Discuss or investigate the generalizability of the learned head weights. Reporting results for weights calibrated on one dataset (e.g., CUB) but applied to another (e.g., ImageNet) would demonstrate if the object/context separation is a general property of specific attention heads. Furthermore, to provide essential context for this analysis, the Abstract and Introduction must clearly state that the CHILI calibration step requires a probing dataset with ground truth segmentation masks for concept localization, as this data dependency affects the zero-shot claim for the overall process.

---

> ### Author Response · Authors · 2025-11-21
> **Response to Reviewer LVKt**
>
> We sincerely thank the reviewer for their positive assessment of our work and for acknowledging its relevance. Below, we address the specific point raised regarding the probing dataset.
> First, we wish to clarify that, in our experiments, the probing dataset is distinct for each dataset. We have revised the manuscript (Section 4.2) to eliminate any ambiguity on this matter.
> To further explore the robustness of our approach, we investigated the impact of applying the weights derived from the ImageNet dataset to perform disentanglement on images from another dataset (MonuMAI) and vice versa. The results of this cross-dataset evaluation are presented in the table below:
> | Probing dataset | Test dataset | Auroc  |
> |-----------------|--------------|--------|
> | Imagnet     	| Monumai  	| 0.557 (0.015) |
> | Monumai     	| Monumai  	| 0.565 (0.015) |
> |-----------------|--------------|--------|
> | Monumai     	| Imagnet  	| 0.589 (0.016)  |
> | Imagnet     	| Imagnet  	| 0.596 (0.017) |
>
> While we observed an expected decrease in performance—attributable to the fact that the probing data originates from a different distribution than the inference set—this decline was modest. We hypothesize that this resilience stems from similarities in image content across datasets. This suggests that the learned representations retain a degree of generalizability despite differences in annotation granularity. We believe this finding underscores the potential robustness of our method and have included this analysis in the revised manuscript.

---

### Review · Reviewer_CTGW · 2025-10-23

**Summary Of Contributions:**

This paper addresses concept hallucination in CLIP-based Concept Bottleneck Models (CBMs), where CLIP incorrectly predicts concept presence based on contextual cues rather than actual visual presence. The authors propose CHILI (Concept Hallucination Inhibition via Localized Interpretability), a post-hoc method that disentangles CLIP activation maps into object-related, context-related, and pseudo-register components without requiring model fine-tuning.

**Key Strengths:**
- HIghlights an important problem in CLIP-based CBMs through statistical analysis across three datasets (ImageNet, MonumAI, CUB)
- Proposes a principled decomposition method.
- Shows improvements in localization tasks (mIoU, mAP)
- Works as a post-hoc method without retraining, making it practical for existing systems

**Key Weaknesses:**
- The performance gains over different baselines are very marginal and may not be statistically significant.
- Table 1 presents a statistical analysis comparing results across three datasets to demonstrate the hallucination problem. However, this same style of analysis is not repeated after introducing CHILI to verify whether the proposed method actually resolves the identified issues.
- The paper provides no justification or evidence for why the median filter is the appropriate choice for removing pseudo-register artifacts, or why this operation specifically removes non-spatial global information.
- The paper suffers from clarity and organization issues. Baselines are not denoted in a standard way, sections could be better structured, and multiple readability problems exist throughout.

**Audience:**

Yes

**Audience Explanation:**

The paper addresses relevant topics for the machine learning community. The problem is timely and important. Concept Bottleneck Models and vision-language foundation models are active research areas, and understanding their failure modes matters for building trustworthy systems. The finding that CLIP hallucinations undermine CBM faithfulness has practical implications for researchers and practitioners.

The methodological contribution, builds on existing attention decomposition techniques, hence offering a useful adaptation to the CBM context with a specific focus on localization versus contextual suggestion.

The experimental design is clear and uses standard datasets, which should facilitate future work in this area.

**Broader Impact Concerns:**

The paper does not include a Broader Impact Statement. While the work on improving interpretability is valuable, the authors should briefly address its limitations in high-stakes applications. Since CHILI only partially mitigates concept hallucination, practitioners should understand the remaining limitations when deploying these methods in sensitive domains like healthcare or legal systems.

**Claims And Evidence:**

No

**Claims Explanation:**

Several claims need stronger support:

- While Table 1 documents the hallucination problem with 40-50% failure rates, Table 2 shows CHILI only achieves AUROC scores of 0.53-0.60, which suggest concept detection with barely above chance probability. This suggests the method only partially resolves the issue, yet the paper doesn't adequately explain this limitation.

- The segmentation results in Table 3 show modest improvements (2-3 points in mIoU), but it's unclear how these metrics directly measure concept hallucination reduction versus general localization ability.

- The claim that CHILI produces "more interpretable CBMs" lacks quantitative support. Section 6.2 provides only qualitative visual examples without user studies or established faithfulness metrics.

- The individual contributions of pseudo-register removal, the weighting scheme, and the hyperparameter alpha are not isolated. Without this, it's difficult to assess which design choices matter.

- No statistical significance tests are provided for any results. Given the small effect sizes in some experiments, it's unclear whether improvements are meaningful or within noise margins.

- The performance improvement claims seem exaggerated and must be toned down.

**Requested Changes:**

### Critical Changes

- **Add ablation studies** (Section 4.2): Separate experiments are needed to isolate contributions from (a) pseudo-register filtering, (b) the weighting scheme, and (c) the hyperparameter alpha. Without this, readers cannot understand which components drive the results.
- **Provide statistical significance tests**: Results of table 2 need confidence intervals or significance tests to demonstrate that improvements are not due to chance variation.
- **Repeat Table 1 style analysis after introducing CHILI**: The statistical analysis format used in Table 1 should be performed on CHILI to verify whether CHILI helps with the hallucination issues identified initially, against the S_context representation and complete representation.
- **Justify median filter choice**: Provide theoretical or empirical justification for why the median filter is appropriate for removing pseudo-register artifacts and why this operation removes non-spatial global information.
- **Improve writing and organization**: Use standard notation for baselines throughout the paper, restructure sections for better flow, and address readability issues.


### Changes that Would Strengthen the Work

- **User study for interpretability**: Human evaluation would validate whether CHILI-based explanations are actually more interpretable.
- **Computational cost analysis**: Report time and memory overhead compared to standard CLIP-based CBMs.

---

> ### Author Response · Authors · 2025-11-21
> **Response to Reviewer CTGW (1/2)**
>
> We sincerely appreciate the reviewers' valuable feedback, which has significantly contributed to improving our manuscript. Below, we address each of the points raised in detail.
>
> ## Performance Gains and Claims
>
> We would like to clarify that our manuscript does not claim to fully resolve the localization issue. Rather, CHILI represents a first contribution toward mitigating this challenging problem , with the hope that future research will build upon and improve our approach. While the performance gains are modest, they remain competitive with state-of-the-art methods for this particularly difficult task. Mitigating concept hallucination—especially with minimal intervention—is inherently challenging, and it is possible that the problem may not be fully solvable under current constraints. We have revised the tone of the manuscript to moderate our claims and provide a more nuanced discussion of these limitations. Concerning limitations, we discussed themes in Section 7, and we added the discussion of some failure cases in Appendix E.
>
>
> ## Comparative Performance Analysis
>
> Here are the results of CHILI as the formalism of Table 1. Note that since the magnitude of the values are different, the results are normalised.
>
> | Dataset  | Base   | c = c1; k in c1 (present) | c = c1; k not in c1 (absent) | Failure rate |
> |----------|--------|---------------------------|-----------------------------|--------------|
> | ImageNet | Baseline    | 0.853 (0.005)         | 0.846 (0.005)           | 0.4          |
> |          | Chili  | 0.713 (0.035)         | 0.668 (0.040)           | 0.2          |
> | MonumAI  | Baseline    | 0.828 (0.025)         | 0.768 (0.021)           | 0.4          |
> |          | Chili  | 0.680 (0.035)         | 0.712 (0.039)           | 0.3          |
> | CUB      | Baseline    | 0.779 (0.029)         | 0.794 (0.030)           | 0.5          |
> |          | Chili  | 0.507 (0.070)         | 0.548 (0.082)           | 0.3          |
>
> The values in parentheses represent standard variances. Consistent with the findings in Section 5, our results indicate that CHILI yields a modest yet measurable improvement, reflected in both the relative score reduction and failure rate decrease.
>
> ## Choice of Artifact Mitigation Methods
>
> To address pseudo-register artifacts, we systematically evaluated multiple approaches: erosion/dilation, Principal Component Analysis (PCA), thresholding, and median filtering. Our decision to adopt median filtering was informed by an ablation study, detailed below, which assessed their impact on object segmentation performance:
>
>
> | Method                          | pixAcc  | mIoU   | mAP   |
> |---------------------------------|---------|--------|-------|
> | Baseline Code                   | 0.7678  | 0.5714 | 0.8289 |
> | + Erosion/Dilation              | 0.7474  | 0.4795 | 0.7070 |
> | + PCA                           | 0.6627  | 0.4503 | 0.7525 |
> | + Thresholding                  | 0.7700  | 0.5775 | 0.8295 |
> | + Median Filtering              | **0.7827**  | **0.5945** | **0.8460** |
>
> Median filtering and thresholding emerged as the most effective methods, likely due to the high-norm, localized nature of the artifacts [1]. We ultimately selected median filtering for its superior performance and to avoid introducing an additional hyperparameter (as required by thresholding).
>
> ## Contribution of Individual Components
>
> We conducted an ablation study to isolate the contributions of filtering and weighting. The results for object segmentation on ImageNet are presented below:
>
>
> | Filter Register | Weighting | pixAcc  | mIoU   | mAP   |
> |-----------------|-----------|---------|--------|-------|
> | FALSE           | FALSE     | 0.7678  | 0.5714 | 0.8289 |
> | TRUE            | FALSE     | 0.7827  | 0.5945 | 0.8460 |
> | FALSE           | TRUE      | 0.7735  | 0.5827 | 0.8333 |
> | TRUE            | TRUE      | **0.7879**  | **0.6022** | **0.8486** |
>
> The results for binary classification, also on ImageNet are presented below:
>
> | Filter Register | Weighting | AUROC  |
> |-----------------|-----------|--------|
> | FALSE           | FALSE     | 0.495  |
> | TRUE            | FALSE     | 0.566  |
> | FALSE           | TRUE      | 0.570  |
> | TRUE            | TRUE      | **0.596**  |
>
> In both cases, each component individually improves performance, with the combined approach yielding the best results.
>
>
> ## On the Choice of Alpha ($\alpha$)
>
> We also analyzed the sensitivity of our method to the hyperparameter $\alpha$, which controls the weighting scheme:
>
> | Alpha | AUROC  |
> |-------|--------|
> | 1     | 0.586  |
> | 5     | 0.582  |
> | 10    | **0.596**  |
> | 50    | 0.587  |
> | 100   | 0.587  |
>
> The optimal value of α = 10 reflects a trade-off: as α approaches 0, weights become insensitive to IoU, while as α approaches infinity, weights approximate an indicator function (IoU > 0).

---

> ### Author Response · Authors · 2025-11-21
> **Response to Reviewer CTGW (2/2)**
>
> ## On Statistical Significance
>
> We have updated Table 2 to include variance estimates for each term, ensuring transparency in our results:
>
>
> |            | Monumai       | ImageNet      | CUB          |
> |------------|---------------|---------------|--------------|
> | LTC        | 0.536 (0.020) | 0.566 (0.016) | 0.532 (0.013) |
> | loc-median | 0.565 (0.015) | 0.596 (0.017) | 0.533 (0.020) |
> | Register   | 0.548 (0.018) | 0.495 (0.039) | 0.487 (0.019) |
> | Base       | 0.516 (0.018) | 0.495 (0.022) | 0.485 (0.014) |
>
> ## On Computational Costs
>
> While CHILI introduces additional computational overhead, the memory cost is negligible since the only additional information is the weight matrix (of size number of layers*number of heads). The architecture remains the same. Concerning inference time, as an illustration, on a V100 GPU, we currently average the following times:
>
> | Method    | Iters / second |
> |-----------|----------------|
> | Baseline  | 51.64          |
> | CHILI     | 1.13           |
>
> While there is a significant increase of inference time, we consider it as reasonable since the architecture allows to precompute CLIP scores in order to train the CBM.
>
>
>
> —
>
> We welcome further suggestions to enhance the clarity and rigor of our manuscript. If the reviewers have specific recommendations for formatting or additional analyses, we are fully committed to implementing them. All addressed points have been incorporated into the revised version.

---

### Review · Reviewer_5ZdL · 2025-11-10

**Summary Of Contributions:**

The paper presents CHILI (Concept Hallucination Inhibition via Localized Interpretability), a method for enhancing concept localization in CLIP-based Concept Bottleneck Models (CBMs). The core idea is to disentangle CLIP’s activations into object-related, context-related, and pseudo-register components, aiming to mitigate concept hallucination and improve interpretability without fine-tuning. Experiments on ImageNet, MonumAI, and CUB datasets demonstrate consistent improvements in localization and segmentation performance, with minimal loss in classification accuracy when integrated into CBMs.

**Audience:**

Yes

**Audience Explanation:**

The paper’s findings address a current and significant gap in explainable AI for vision-language models, a topic that aligns directly with TMLR’s readership. Several reasons support this:

1. Relevance to Concept Bottleneck Models (CBMs): CBMs are a central theme in interpretable learning, and the paper extends them to the CLIP setting, which is widely studied. TMLR readers include researchers in interpretability, representation learning, and multimodal AI.

2. Methodological contribution (CHILI): The proposed disentanglement of CLIP activations into object, context, and pseudo-register components provides a new direction for mitigating hallucination without retraining. Such post-hoc interpretability approaches are of particular interest to TMLR readers focusing on practical, model-agnostic explainability.

3. Empirical insight: The quantitative evaluation across ImageNet, MonumAI, and CUB datasets gives concrete evidence of concept hallucination and its mitigation, which are empirical findings that would interest both application-driven and theoretical researchers.

**Broader Impact Concerns:**

No.

**Claims And Evidence:**

Yes

**Claims Explanation:**

The authors provided good technical details introduction and comprehensive experimental analysis. Specifically:

1. The problem of concept hallucination in CLIP-based CBMs is clearly motivated and relevant for improving the faithfulness of explainable AI.
2. The method is analytically grounded, decomposing CLIP activations at the layer/head level and proposing a principled separation of object and context signals.
3. Empirical evaluation is extensive across multiple datasets and tasks, showing consistent interpretability gains.

**Requested Changes:**

1. The claim regarding pseudo-register tokens (page 7, Section 4.2) lacks analytical or empirical justification. The paper states that “these tokens act like pseudo-registers: they tend to store global information, but their spatial localization on the activation map is not meaningful for interpretability,” and that this component is separated accordingly. However, no quantitative or qualitative evidence is provided to validate this behavior in the CLIP setting. Since this assumption is critical to the decomposition, the authors should provide supporting analysis or visualization demonstrating the behavior of these tokens.

2. Minor typographical and formatting issues should be corrected. For example,  in Table 4, inconsistent commas and decimal separators appear in numerical values (e.g., “73,55” instead of “73.55”). These errors reduce polish and readability.

3. The qualitative results (Figures 3) only present positive examples where CHILI explanations align well with human perception. For a more convincing interpretability assessment, the paper should also include negative samples, cases where the predicted concept or heatmap has little or no visual correspondence with the image. Showing such counterexamples would clarify the method’s failure modes, illustrate the limits of disentanglement, and help readers assess robustness and interpretive reliability under less ideal conditions.

---

> ### Author Response · Authors · 2025-11-21
> **Response to Reviewer 5ZdL**
>
> We sincerely thank the reviewers for their thoughtful and constructive feedback on our manuscript. We are pleased that you share the importance of addressing concept hallucination in Concept-Based Models. Next, we answer the points of discussion mentioned.
> ## Clarification on Pseudo-Registers
> While our work focused on identifying and filtering high-norm token artifacts, the attribution of these artifacts to registers originates from the findings of [1]. The authors demonstrate that incorporating registers into the model architecture effectively mitigates these artifacts. Given this empirical support, we adopted the term "pseudo-registers" for clarity and convenience in our discussion. We acknowledge that this terminology may have caused confusion and have revised the manuscript to provide further clarification on this point (Section 4.2).
> ## Addressing Typographical Errors
> We are grateful to the reviewers for identifying typographical errors in our submission. These have been carefully corrected in the revised version of the manuscript.
> ## Discussion of Failure Cases
> The appendix E of our manuscript includes illustrative examples of CHILI’s limitations. In short, a recurrent issue we observed is the tendency of heatmaps to overfocus on specific regions of the image—such as the heads of birds in the CUB dataset—resulting in the mislocalization of certain features. We hypothesize that this behavior stems from the use of a uniform weighting scheme across all concepts. Additionally, while median filtering generally performs well, it can occasionally struggle with small concepts, though such cases are rare in practice. We also observed that our method faces challenges with shape-related concepts. To address these concerns, we have expanded our analysis of failure cases in the revised manuscript in appendix E, offering a more comprehensive discussion of these limitations and their implications.
>
>
> [1] Timothée Darcet, Maxime Oquab, Julien Mairal, and Piotr Bojanowski. Vision transformers need registers. In The Twelfth International Conference on Learning Representations, 2024.

---

### Decision · Action_Editor_fvZn · 2025-12-15

**Recommendation:** Accept as is

**Audience:**

Yes

**Audience Explanation:**

Yes, a substantial audience in the TMLR community would find this work to be relevant and interesting.

**Claims And Evidence:**

Yes

**Claims Explanation:**

Yes. After the author-reviewer discussions and the revision, all three reviewers agree that (most of) the submission's claims are supported by accurate, convincing and clear evidence. Reviewer CTGW asked for additional discussion about the performance improvement due to reduced hallucination versus overall better localization, as well as quantitative support for better interpretability. Nonetheless, the reviewer found most of their concerns addressed by the rebuttal, and is in favor of acceptance. The AE therefore believes that the submission is ready to be published at TMLR.